# The Birthplace Effect in 14–18-Year-Old Athletes Participating in Competitive Individual and Team Sports

**DOI:** 10.3390/sports10040059

**Published:** 2022-04-11

**Authors:** Zohar Maayan, Ronnie Lidor, Michal Arnon

**Affiliations:** Wingate Institute, The Academic College at Wingate, Netanya 4290200, Israel; zoharm@wincol.ac.il (Z.M.); michalar2@gmail.com (M.A.)

**Keywords:** birthplace, individual sports, team sports, youth sport programs, sport policy

## Abstract

The birthplace (the place where an athlete was born) effect (BPE) has been found to be one of the environmental variables associated with early talent development and the achievement of a high level of proficiency in sport. The purpose of the current study is twofold: (1) to calculate the BPE in 14–18-year-old athletes who participated in individual and team sports and (2) examine how coaches perceived this effect. The participants were 1397 athletes (390 females and 1007 males) who competed in 5 individual (gymnastics, judo, swimming, tennis, and track and field) and 5 team (basketball, soccer, team handball, volleyball, and water polo) sports, as well as 147 coaches who provided their preliminary thoughts about the BPE. Data analyses revealed that although the BPE was not found to be associated with cities of a similar size, it was observed that growing up in cities of small and medium sizes was more beneficial than growing up in towns or cities of other sizes. Most of the coaches believed that certain characteristics of the place or city where the athlete grew up (e.g., proximity to sport facilities) could contribute positively to the athlete’s development. We discuss how the BPE data can aid policymakers in developing a sport policy associated with early phases of talent development.

## 1. Introduction

One of the environmental factors linked to the development of expertise in sport is the place where an athlete was born (i.e., the birthplace effect (BPE)) (see, for example, [1,2]). A typical quantitative study on the BPE examines the association between the distribution of athletes’ city of development (determined by birthplace or city of registration) within the sport program or system and the distribution within the general population using census data (e.g., [1,2,3,4]). The BPE was examined in a series of studies on both male and female athletes in various individual and team sports (see, for example, [3,5]).

BPE data were collected on male athletes in major sports, among them being baseball, basketball, golf, hockey [2], football [6], and junior ice hockey [7]. In addition, BPE data were obtained from female elite performers in soccer and golf [8]. The findings from these studies indicated that male and female athletes who were born in cities of small-to-medium sizes were more likely to play for professional leagues and attain a higher level of sport participation than athletes who were born in larger cities. In other words, athletes who were born in cities of a small or medium size had a greater chance of reaching the highest level in their given sports.

Support for the above-mentioned BPE findings was also found in quantitative studies using an alternative approach to examining this effect: population density. In one study conducted on male Danish team handball and soccer players, it was reported that the overall participation rates were higher among players who were born in low-density communities rather than high-density communities [9]. In another study on Portuguese volleyball players, it was observed that the birthplace population density was significantly lower for players who played for First League teams (the highest level of soccer competition in the country) than players who played for Third League (a lower level of competition) teams [10]. 

A number of arguments has been made to explain why it was more beneficial for young athletes to grow up in cities of a small-to-medium size than in cities of a different size. Although large cities can provide children and youths with enhanced conditions such as well-designed and equipped sporting facilities, as well as better coaching guidance, it appears that the large cities’ sport programs are heavily structured around and hindered by the lack of space and time in which young athletes can participate [11,12]. The physical environments of smaller cities or towns allow for instructional and social benefits that cannot be gained in larger cities, among them being (1) a greater amount of independent mobility and physical safety, (2) an integrative approach to sport participation involving schools, families, and the community at large, and (3) a more personal relationship between athletes and coaches. As noted by MacDonald, King, and their colleagues [8], “… the developmental opportunities for nurturing sporting talent offered by small towns and cities may be somehow superior to the development opportunities of larger cities” (p. 234).

Similar explanations for the benefits athletes can gain in cities of small-to-medium sizes were also provided in a number of qualitative studies [13,14]. In a case study, Balish and Côté [13] performed interviews and analyzed documents in order to explore how a small and successful sport community (646 residents) contributed to the development of the local athletes. Three themes emerged from this study: (1) developmental experiences concerning youths engaging in organized and unorganized sport activities where teammates remained stable throughout development, (2) community influences concerning the interdependence between the local schools and the community, and (3) sociocultural influences concerning youths who possess a collective identity coupled with an intense inter-community rivalry.

Of particular interest to the aims of the current BPE study are two studies in which BPE data were collected on elite male and female Israeli athletes. In one study [15], data were obtained from 521 male ball game players who played in Division 1 (the highest division for competitive ball games in Israel). Mixed findings for the BPE across sports were found. For the soccer players, it was indicated that the likelihood of players who were born in very small communities or a small city to play for teams in Division 1 was lower than for players who were born in a city of a different size. In addition, the likelihood of soccer players who were born in a medium-sized city to play in Division 1 was higher than that for players who were born in a town or city of another size.

Growing up in a city of a medium size was also found to be advantageous for team handball players. The likelihood of players who were born in a medium city to play in Division 1 was higher than that for players who were born in a city of a different size. For the volleyball players, it was indicated that the likelihood of players who were born in a small place (<2000 people) to play in Division 1 was higher than that for players who were born in a city of a different size. The main finding of Lidor et al.’s [15] study revealed that the likelihood of reaching the highest level of competition in team sports, such as soccer and team handball, was higher among those who were born in cities of a medium size than among those who were born in cities or towns of a different size.

In another study on female elite athletes [16], BPE data were collected on 389 ball game individuals. The main findings of this study were that for the team handball players, the likelihood for players who were born in a medium-sized city to play in Division 1 was higher than that for players who were born in a city of a different size. In addition, for the volleyball players, it was found that the likelihood of players who were born in a small place (<2000 people) to play in Division 1 was higher than that for players who were born in a place of a different size. These volleyball data were similar to the volleyball data obtained in Lidor et al.’s [15] study. Finally, for the basketball players, team handball players, and volleyball players, the likelihood of players who were born in a small city to play for teams in Division 1 was lower than that for players who were born in a city of a larger size.

Our main aim in the current study was to assess the BPE in young Israeli athletes (ages: 14–18 years) who participated in competitive sport programs. We selected 14–18-year-old athletes because this age category represents two important periods in the early careers of young competitive athletes. According to Côté’s *Developmental Model of Sport Participation* (DMSP) (see [2,17]), the ages of 13–15 are considered to be the turning point from the developmental years to the specialization years. The age of 18 years is considered to represent the beginning of a transitional phase from being part of a youth sport program to becoming a member of an elite adult sport program in most programs in Israel [15,18]. 

It has already been argued that evidence-based quantitative and qualitative BPE data can assist those professionals who work with young athletes, such as policymakers, program directors, coaches, instructors, and sport psychology consultants, in developing sport policies that focus on positive and enjoyable experiences for children in the early phases of talent development (see [5]). In the current BPE study, we collected data on children and youths who had been part of competitive sport programs for a number of years but had not yet reached the highest level of competition in the local sport structure. Since Israel is considered to be a small nation (see [19]). where only a small portion of the children and youths are involved in competitive individual and team sport programs [18,20], evidence-based data on the size of the place of birth of the young athlete can help those professionals who work regularly with them to increase their understanding of how to recruit children to sports, as well as how to motivate them to maintain their participation in the selected sport program(s) for longer periods of time. In line with previous findings on the BPE (e.g., [7,8,15,16]), we assumed in the current study that the benefits associated with growing up in cities of a medium size would result in a high distribution of 14–18-year-old athletes who participated in sport programs compared with their distribution within the general population.

In order to enable the analysis of long-term developmental trends that are related to competitive sport programs in a given city or country, data on core environmental factors (e.g., the BPE) should be collected regularly and analyzed across different phases of development [18,21]. In the current study, we compared the BPE of young athletes in competitive sports programs with BPE data collected in two previous studies [15,16]. The aim was to conduct a further examination of the BPE in 14–18-year-old individuals over a 10-year period.

To complement the BPE data collected on the young athletes, we also attempted to explore how coaches who worked with children and youths perceived the effect. More specifically, we aimed at examining how coaches valued the contribution of the city or place where the athletes grew up to their development. In order to conceptually understand the contribution of BPE to the development of young athletes, it might be beneficial to collect data on how professionals who work regularly with these athletes perceive the existence or nonexistence of the effect, such as coaches who play a major role in the long-term developmental pathways of the athletes (see [22,23]). Such qualitative data could provide additional insights into the BPE.

As such, the purpose of the current study was twofold: (1) to examine the contribution of Israeli athletes’ (ages 14–18 years) city sizes (i.e., population) to sports participation and performance and (2) collect preliminary data on how the coaches perceived this effect. The data on the BPE of the young athletes obtained in the current study were compared with the BPE data collected in two previous studies on elite male [15] and female [16] Israeli ballplayers and to examine specific BPE trends in one local sport structure from a perspective of approximately one decade.

## 2. Methods

### 2.1. Participants

The BPE was assessed in 1397 14–18-year-old athletes (mean age = 16.37 years; *SD* = 1.54), among them being 390 females (mean age = 16.28 years; *SD* = 1.57) and 1007 males (mean age = 16.40 years; *SD* = 1.53). The athletes were part of 10 sports: 5 individual (gymnastics, judo, swimming, tennis, and track and field) and 5 team (basketball, soccer, team handball, volleyball, and water polo) sports. The number of athletes in each sport is described in Table 1. The low number of females who took part in our study demonstrates the low number of young active female athletes in the country. For example, according to the Central Bureau of Statistics, only 19% of the active athletes 13–18 years of age in Israel are females (see [24]). We were aware not only of the low number of female athletes who participated in the current study but also the low number of male athletes in some of the sports (e.g., only 18 male gymnasts took part in the study). Our findings relating to the BPE are presented while taking into account the restrictions of the relatively small sample size. In a previous study, we used this sample of young athletes to calculate their relative age effect [18]. 

The participants in the current study were recruited from leading individual and team sport programs. We did not recruit athletes from all existing sport programs in the country to ensure that the young participants in the current study were all members of well-organized sport programs; that is to say, the female and male athletes in this study participated in between four and six sessions of practice on a weekly basis. The majority of the sport programs in Israel in both individual and team sports permit registration at the age of eight, and therefore, the participants in our study who were aged 14–18 years had at least 6 years of experience in training and competition. In a number of sports, such as gymnastics, swimming, and soccer, children can enter the sport program even earlier than the age of eight, and thus, some of them had undergone at least 9 years of training and competition. 

**Coaches.** In addition to the players, 147 coaches (82 of individual sports and 65 of team sports; mean coaching experience = 14.20 years; median = 12 years) participated in the study. Among these coaches, 68 (14 females and 54 males) worked directly with the 14–18-year-old athletes who participated in our study, and 79 worked in the same sports programs as the other coaches but did not work with the athletes who participated in the current study. All the coaches were certified by their sports federations.

### 2.2. Procedure

This study was approved by the ethics committee of the Academic College at Wingate (Ethic Code # 112).

Information about the athletes’ birthplaces, genders, and types of sport was obtained via questionnaires, which were given to the athletes by their coaches. Each director of the sport program where the athletes who participated in the study practiced received a letter providing the background and objectives of this study. In addition, informed consent was obtained from the parents of the participants. After approval was obtained, the first author approached the coaches of the athletes (*n* = 68) and sent them the questionnaires via electronic mail.

To examine the BPE in the 14–18-year-old athletes participating in our study, similar procedures to the ones performed in previous studies (e.g., [2,16]) were implemented. The city size of the BPE of each athlete from each of the 10 sports was based on Israel Census data obtained from a demographics website (see www.cbs.gov.il/shnaton60/st02_11x.pdf (accessed on 10 February 2022)). Four categories of city size were used in the study, as performed in previous BPE studies on Israeli athletes (for more details, see [15,16]). The designation of these four categories allowed us to examine the BPE in four sizes of residential areas: small towns and small, medium, and large cities.

The selection procedure we used in the current study was also applied in two previous studies on Israeli elite male [15] and female [16] ball game players, which therefore allowed us to compare the data obtained from the same categories of city size across a period of about a decade. It should be noted that information was unavailable on the migration between large cities and small cities or between small cities and large cities. It was assumed that the net movements between large cities and small cities or between small cities and large cities were likely to be equal.

Since the mean age of the participants (females and males) was 16.32 years and the majority of the data in our study were collected in 2016, we used the census data presented in 2014. That year was the most appropriate one to reflect the ages of 14 of the athletes (i.e., a turning point from developmental years to specialization years) who took part in our study.

In order to collect preliminary information on how the BPE is perceived by coaches, 150 coaches were approached by the first author. The coaches were asked to answer in writing a closed question related to the BPE: “Do you think that the athletes’ places of birth affect their development?” There were two options for the answer: “yes” or “no”. Only if the answer was “yes” was the coach asked another (open) question: “How does the place of birth contribute to the athletes’ development?” The coaches were asked to outline how the BPE contributed to the athletes’ development. Out of the 150 coaches who were approached, 147 responded (rate of responsiveness = 98%).

### 2.3. Data Analysis

To test the BPE, odds ratios (ORs) were calculated to determine the likelihood of participating in each of the sport programs (compared to the distribution of the population at age 14; year 2014) for each city size, and 95% confidence intervals (CIs) were calculated around each OR. An OR greater than 1 (with upper and lower limits higher than 1) implied that an athlete born in the given city size was more likely to become a participant in the sport program in the given sport than if they had been born in any other city size. An OR of less than 1 (with upper and lower limits less than 1) implied that an athlete born in the given city size was less likely to become a participant in the sport program than if they had been born in a city of a different size. ORs including the value of 1 within their CI range were not considered to be statistically significant.

As indicated previously, we were aware of the fact that the sample size of the female athletes was relatively small (390 participants). However, it was our aim to analyze their data across the different sports in order to strengthen our understanding of the existence of the BPE in the various sport programs available to female children and youths in the country. The responses of the 147 coaches who provided exploratory information about the BPE were descriptively analyzed. The coaches’ responses were also thematically analyzed.

## 3. Results

The results are presented separately for the BPE and the responses of the coaches.

### 3.1. BPE Data

The representations of the Israeli population aged 14–18 years, the male athletes, and ORs and CIs across cities of different sizes are presented in Table 2 (individual sports) and Table 3 (team sports). The information on the female athletes who participated in our study is presented in Table 4 (individual sports) and Table 5 (team sports). Mixed results were found for the BPE in the male and female athletes and across the 10 sports.

#### 3.1.1. Male Athletes

Data are presented separately for individual sports and team sports.

##### Individual Sports

In only two sports was the BPE found to be significant: gymnastics and swimming. In gymnastics, it was found that small towns of up to 2000 people yielded an OR significantly higher than 1. This means that the likelihood of participating in a competitive sport program for those gymnasts who were born in a small place was higher than that for those who were born in a city of a different size. In addition, cities with a population of 50,000–200,000 yielded an OR significantly lower than 1. The likelihood for gymnasts who were born in a medium-sized city to be part of a competitive sport program was lower than that for those who were born in a city with a smaller or larger population.

Similar results were obtained in swimming; small towns of up to 2000 people yielded an OR significantly higher than 1. This means that the likelihood of participating in a competitive swimming program for swimmers who were born in a small place was higher than that for those who were born in a town or city of a different size. In addition, cities with a population of 50,000–200,000 yielded an OR significantly lower than 1. The likelihood for swimmers who were born in a medium city to be part of a swimming program was lower than that for those who were born in a city with a smaller or larger population.

##### Team Sports

The BPE was found to be significant in each sport; however, mixed results were observed. In basketball, cities with a population of 50,000–200,000 yielded an OR significantly higher than 1. This means that the likelihood for basketball players who were born in a medium-sized city to be part of a basketball program was higher than that for those who were born in a city of a larger or smaller size. In soccer, the findings for cities of the same size (50,000–200,000) were found to be the opposite; cities of this population size yielded an OR significantly lower than 1, and the likelihood for the soccer player who was born in a medium-sized city to participate in a competitive soccer program was lower than that for those who were born in a city with a smaller or larger population.

For the team handball players, cities with a population of 50,000–200,000 yielded an OR significantly higher than 1. This means that the likelihood for team handball players who were born in a medium-sized city to be part of a team handball program was higher than that for those who were born in a city of a larger or smaller size. It was also found that small towns of up to 2000 people yielded an OR significantly lower than 1. The likelihood of participating in a competitive sport program for those team handball players who were born in a small place was lower than that for those who were born in a place with a higher population.

For the volleyball players, small towns of up to 2000 people yielded an OR significantly lower than 1. This means that the likelihood of participating in a competitive sport program for those volleyball players who were born in a small place was lower than that for those who were born in a city of a different size. Finally, for the water polo players, cities with a population of more than 200,000 people yielded an OR significantly higher than 1. This means that the likelihood for water polo players who were born in a big city to be part of a water polo program was higher than that for those who were born in a city of a different size.

#### 3.1.2. Female Athletes

Data are presented separately for individual sports and team sports.

##### Individual Sports

The BPE was found to be significant in three sports: gymnastics, judo, and track and field. For the gymnasts, cities with a population of 2000–50,000 or 50,000–200,000 people yielded an OR significantly lower than 1. This means that the likelihood for gymnasts who were born in small-to-medium-sized cities to be part of a competitive sport program was lower than that for those who were born in a city of a different size. For the judokas, it was found that small towns of up to 2000 people yielded an OR significantly higher than 1. This means that the likelihood of participating in a competitive judo program for those individuals who were born in a small place was higher than that for those who were born in a city with a larger population.

For the track and field athletes, being born in a small town was also found to be beneficial; small towns of up to 2000 people yielded an OR significantly higher than 1. This means that the likelihood of participating in a competitive sport program for those track and field athletes who were born in a small place was higher than that for those who were born in a city with a larger population. In addition, it was found that cities with a population of 50,000–200,000 people yielded an OR significantly lower than 1. The likelihood for those who were born in medium-sized cities to be part of a competitive sport program was lower than that for those who were born in a city of a different size.

##### Team Sports

The BPE was found to be significant in all team sports but water polo. (The number of the active players was very small, and therefore, BPE analyses were not performed). Mixed results were indicated. For the basketball players, cities with a population of more than 200,000 people yielded an OR significantly higher than 1. This means that the likelihood for basketball players who were born in a big city to be part of a basketball program was higher than that for those who were born in a city with a smaller population. In addition, cities with a population of 2000–50,000 people yielded an OR significantly lower than 1. This means that the likelihood for those who were born in small cities to be part of a competitive sport program in basketball was lower than that for those who were born in a city of a different size.

For the soccer players, cities with a population of 50,000–200,000 yielded an OR significantly lower than 1. The likelihood for soccer players who were born in a medium-sized city to be part of a competitive sport program was lower than that for those who were born in a city of a different size.

For the team handball players, cities with a population of 50,000–200,000 people or more than 200,000 people yielded an OR significantly higher than 1. The likelihood for team handball players who were born in medium- and large-sized cities to be part of a competitive sport program was higher than that for those who were born in a city of a smaller size. In addition, cities with a population of 2000–50,000 people yielded an OR significantly lower than 1. This means that the likelihood for those who were born in small cities to be part of a competitive team-handball program was lower than that for those who were born in a city of a different size.

For the volleyball players, small towns of up to 2000 people yielded an OR significantly higher than 1. This means that the likelihood to participate in a competitive volleyball program for those who were born in a small place was higher than that for those who were born in a place with a higher population.

### 3.2. Coaches’ Preliminary Thoughts on the BPE

The coaches’ responses associated with the BPE are presented in Table 6. More than 80% of the coaches (about 85% of the coaches who worked with individual sport athletes and 81% of those who coached team sport players) answered “yes” when they were asked if the athletes’ places of birth contributed to their development. Three main observations associated with the BPE emerged: (1) proximity to sport facilities, where the coaches argued that living near sports facilities is beneficial for the young athlete. Among the benefits they outlined were that the athletes spent less time on the roads, there was no need for adults to transport them to practices, and the athletes had a greater feeling of security; (2) the socioeconomic statues of the living place, where the coaches claimed that wealthy places or cities can provide better instructional support and more formal and informal learning opportunities for athletes to develop their athletic abilities and skills than what less wealthy towns or cities can offer; and (3) sport popularity in a given place, where the coaches argued that it is easier for children to select a given sport if they grow up in a town or city where the sport is popular and supported by the community.

## 4. Discussion

The discussion is composed of three parts. In the first part, we discuss the BPE data from the 14–18-year-old athletes, and in the second part, we examine the reflections of the coaches on the contribution of the place of residence to the young athletes’ development. Based on the BPE data obtained in our study, in the third part, we elaborate upon a number of aspects associated with the establishment of a national sport policy for talent detection and early phases of talent development. 

### 4.1. BPE Data

Mixed BPE results were found for the male and female athletes across the individual and team sports. The main BPE findings in the male athletes were that (1) for those who participated in individual sport programs—gymnastics and swimming—growing up in a small place (less than 2000 people) had greater benefits than any other sized place of residence; (2) for the athletes who were part of competitive basketball and team handball programs, it was more beneficial to grow up in cities of a medium size (50,000–200,000 people) than in cities of other sizes; and (3) only for the water polo players was it more beneficial to grow up in a large city (more than 200,000 people).

The main BPE findings in the female athletes were that (1) for those who participated in individual sport programs—gymnastics, judo, and track and field—it was more beneficial to grow up in a small place or in a city of a medium size (gymnastics) than in a place of a different size, and (2) for the athletes who played team sports—basketball and team handball—it was more beneficial to grow up in a large city than in a city of any other size. For the team handball players, it was also beneficial to grow up in a medium-sized city. Living in a small place was found to beneficial for only the volleyball players.

The mixed BPE results of our study on the 14–18-year-old athletes were similar to the ones reported in Lidor et al.’s studies [15,16] on elite athletes; that is, no clear-cut observations could be made for the contribution of one city of one size to the development of the athletes across sports. Different cities of different sizes were found to be associated with achieving a high level of proficiency in the 20 (10 for the males and 10 for the females) analyzed sport programs. However, in 12 out of the 20 programs, it was found that growing up in a very small place or a city of a medium size had greater benefits. Living in a city of a large size was indicated to be beneficial to athletes in only three sport programs. In only 7 out of the 20 sport programs was it found to be detrimental to grow up in cities of small or medium sizes.

Therefore, the contribution of small places and cities of a medium size to the development of the young athletes in our study can presumably be explained by the instructional and social characteristics of these places, as discussed in previous BPE studies (e.g., [6,13,16]). Towns and cities of these particular sizes may be more likely to encourage athletes and parents toward certain sports and may have unique resources, such as the availability of sport facilities, that can enhance athletic development (see, for example, [3,5,6,25]). Therefore, the contribution of growing up in small- or medium-sized towns up until the age of 14 to the development of young athletes can be observed not only in large-population countries (e.g., Canada, Germany, or the USA) but also in small countries such as Israel.

From a 10-year perspective, it appears that in Israel, it has been more beneficial for athletes to grow up in towns and cities of a small-to-medium size [15,16]. Even in soccer, which is considered the most popular sport in Israel [20], growing up in large cities was not found to be beneficial for young soccer players. Although the best professional soccer clubs in Israel are situated in large cities (i.e., Haifa, Jerusalem, and Tel-Aviv), our data show that joining a soccer program in a large city may not provide the optimal instructional-psychological conditions needed for attaining proficiency in the game of soccer.

However, the explanations favoring growing up in places or cities of small or medium sizes may not tell the whole story of the findings obtained in our study, since we also found that athletes who participated in 7 sport programs (out of the 20 programs) did not benefit from living in places or cities of small or medium sizes. It appears that additional characteristics of the place or city, rather than only its size, may be associated with the athletes’ development. The analysis of the coaches’ BPE reflections can add another dimension to the quantitatively analyzed BPE data in our study.

### 4.2. Coaches’ Reflections

Most of the coaches pointed out that the place of birth did in fact contribute to the athletes’ development; however, they did not focus on the size of the city but rather on other characteristics. A number of their reflections indeed strengthened the finding that it is more beneficial to grow up in small- and medium-sized cities than in cities of a greater size. For example, the coaches emphasized ways that athletes can benefit from living close to the sports facilities: they can save time on the roads, arrive independently to the facility, and feel secure on their way to and from practice. These observations are typically associated with small towns and cities of a medium size and therefore provide support for the benefits of such places to a young athlete’s development (see, e.g., [5,25]). 

However, the coaches also mentioned that the socioeconomic status of the place or city where the athlete grew up had the potential to influence his or her development. More specifically, they claimed that children and youths are provided with both more formal and informal opportunities to develop their sport skills in wealthy places or cities rather than those available to them in poorer places or cities. The coaches did not relate to cities of a given size. In addition, they did not mention a specific sport program that might be negatively or positively influenced by the socioeconomic status of the place or city where the athlete lived. These thoughts of the coaches reflect the idea that certain conditions, among them the availability of sport facilities, instructional support, and social support (typically associated with wealthy places or cities), resulted in a higher rate of sport participation among children [26]. In addition, prior research (e.g., [27,28]) highlighted the consistency of the socioeconomic status limitations to participation in high-performance sports.

Another perspective that was shared by the coaches was the one related to how popular a sport program was in a given city. The coaches felt that children are initially attracted to popular sport activities; the more popular the sport, the more chances there are that the children will select it for themselves. For example, it was found in our study that the likelihood of participating in competitive sport programs for the female volleyball players who were born in a small place (up to 2000 people) where the game was popular was higher than that for those who were born in a city of a different size. These data can be explained by the fact that in Israel, the game of volleyball was originally played in places like the Kibbutz (cooperative farming settlements) and small towns [18], and therefore, the game of volleyball has been popular in such places.

The combined qualitative reports given by the coaches and the quantitative BPE analysis provide further support for the notion that small-to-medium-sized towns and cities, rather than large cities, are considered to be more suitable sports environments for children and youths (aged 14–18 years old, females and males) to develop their sports skills. The coaches felt that urban conditions, such as living in close proximity to the sports facilities, and the socioeconomic status of the city are detrimental contributors to a young athlete’s early development phases. The coaches argued that such conditions can help young athletes focus on the training program.

Finally, the observations made by the coaches in our study can add dimensions to the concept of BPE, other than the size of the city or the population density. As noted by Wattie et al. [25], “… further research is needed in this area to understand contextual differences related to birthplace” (p. 378). Obtaining data on specific cultural and social characteristics of places or cities of different sizes can increase the understanding of possible mechanisms involved in the association between the BPE and achieving success in sports.

### 4.3. BPE and the Establishment of a (National) Sports Policy in a Small Nation

Up to now, a national sports policy associated with early phases of talent identification and development has not been established in Israel. In one study evaluating a number of aspects of the elite sports policy in Israel [29], only one third of the directors of the elite sports federations who participated in the study claimed that their federation had a well-developed policy for early development in sports. In addition, according to the directors of these federations, there is a lack of support and involvement by sports scientists, multidimensional support services appropriate to the age and level of young athletes, and nationally coordinated support for the combination of sports development and scientific study. De Bosscher and her colleagues summarized that “… priority sports need to include a long-term talent development training plan” (p. 154).

Based on the data of the current study, as well as on similar data collected in previous BPE studies on adult Israeli athletes [15,16] and on athletes who grew up in small communities in other countries [13,14], we discussed two issues associated with the BPE and the development of sports policy for young athletes: how to increase the number of children in individual and team sport programs and the prioritization of sport programs.

In order to increase the number of children who choose to participate in a sport program, it is proposed to sport policymakers that they further develop sport programs in individual and team sports in small- and medium-sized cities by increasing (1) the number of sports facilities available to the public at large and (2) the subsidization of children’s participation in the sport program(s). The public investment in sport program(s) in small- and medium-sized cities may not only benefit the young members of the local community but also those children who live in a large city which is located near the small community. In many areas in Israel, the geographical distance between small communities and large cities is small (e.g., 2–3 km), and therefore, those who live in a large city (e.g., more than 200,000 people) may also benefit from the sport programs taking place in a nearby small- or medium-sized city.

BPE data can also be used by policymakers when decisions about prioritization of sport programs are made (see, for example, [30]). For example, Israeli judokas (females and males) have won medals in several Olympic Games (e.g., in the 2016 and 2020 Games) and World Championships (e.g., in the year 2019). Up to now, Israeli athletes have won 13 Olympic medals, 6 of them by judokas (5 individual medals and 1 team medal). In fact, judo has become the individual sport in Israel with the most achievements at the Olympic level. Looking at the data of our BPE study, it was indicated that only for the female judokas is the likelihood of participating in a competitive judo program for those who were born in a small place higher than that for those who were born in a city of a larger size. If the goal of the Ministry of Culture and Sports and the Judo Federation is to attract more children and youths—not only females but also males—to join judo programs in light of the international success achieved by Israeli elite judokas, then it might be of relevance to develop more judo programs in small- and medium-sized cities. It is assumed that the greater investment in judo programs in cities of these sizes, the higher the potential to increase the popularity of the sport in these communities, and subsequently the higher the chances of increasing the number of active young judokas.

In summary, mixed BPE findings were obtained from our study. No clear-cut conclusions can be made on the contribution of one city of one size to the development of athletes across the observed individual and team sports; that is to say, different sizes of cities were indicated to be linked to attaining success in individual and team sports.

### 4.4. Practical Implications for Policymakers

It might be of interest to policymakers in Israel to develop a national sports policy in order to maximize the potential to recruit children to sport programs and provide them with the appropriate conditions to achieve success. If this is the case, knowing about the contribution of the BPE to an athlete’s success can help policymakers implement these evidence-based data in the decision-making processes. However, if such a national sport policy is not the main interest of the ministry due to its motivation to enable the different sport federations to establish their own sport policies, then the data of the BPE can be used by the federations as well.

## Figures and Tables

**Table 1 sports-10-00059-t001:** The number of male and female athletes across sports.

Sport	Male Athletes	Female Athletes
*Individual Sports*		
Gymnastics	18	46
Judo	175	51
Swimming	104	50
Tennis	127	52
Track and field	58	27
*Team Sports*		
Basketball	158	76
Soccer	215	21
Team handball	51	35
Volleyball	66	29
Water polo	35	3
Total	1007	390

**Table 2 sports-10-00059-t002:** Representation of the Israeli population, male individual sport athletes ^a^, and ORs and CIs across cities of different sizes.

City Size	Israel Pop ^b^	Gymnastics			Judo			Swimming			Tennis			Track and Field		
		%	OR	CI	%	OR	CI	%	OR	CI	%	OR	CI	%	OR	CI
>2000	8.57	27.78	4.10	(1.80, 9.37) *	7.39	0.85	(0.31, 2.37)	25.96	3.74	(1.63, 8.59) *	14.17	1.76	(0.72, 4.32)	13.79	1.71	(0.69, 4.21)
2000–50,000	34.9	33.33	0.93	(0.52, 1.67)	38.07	1.15	(0.64, 2.04)	31.73	0.87	(0.48, 1.56)	33.07	0.92	(0.51, 1.66)	31.03	0.84	(0.47, 1.51)
50,000–200,000	29.48	5.56	0.14	(0.05, 0.37) *	24.43	0.77	(0.41, 1.45)	17.31	0.50	(0.26, 0.98) *	24.41	0.77	(0.41, 1.45)	20.69	0.62	(0.33, 1.19)
<200,000	27.06	33.33	1.35	(0.74, 2.47)	30.11	1.16	(0.63, 2.15)	25.00	0.90	(0.48, 1.69)	28.35	1.07	(0.57, 1.98)	34.48	1.42	(0.78, 2.60)

Note: OR = odds ratio; CI = confidence interval. * Significant difference. ^a^ Percentage of male athletes who participated in individual sport programs in 2016 and who grew up in each of the subdivisions of the 2014 census. ^b^ Israel population = percentage of males under the age of 14 in each of the subdivisions of the 2014 Israel Census.

**Table 3 sports-10-00059-t003:** Representation of the Israeli population, male team sport players ^a^, and ORs and CIs across cities of different sizes.

City Size	Israel Pop ^b^	Basketball			Soccer			Team Handball			Volleyball			Water Polo		
		%	OR	CI	%	OR	CI	%	OR	CI	%	OR	CI	%	OR	CI
>2000	8.57	7.55	0.87	(0.31, 2.42)	15.74	1.99	(0.82, 4.82)	0.00	0.00	-	1.52	0.16	(0.03, 0.95) *	14.29	1.78	(0.73, 4.36)
2000–50,000	34.9	25.16	0.63	(0.34, 1.15)	46.76	1.64	(0.93, 2.89)	37.25	1.11	(0.62, 1.97)	27.27	0.70	(0.38, 1.28)	0.00	0.00	–
50,000–200,000	29.48	44.65	1.93	(1.08, 3.46) *	13.89	0.39	(0.19, 0.79) *	50.98	2.49	(1.39, 4.45) *	42.42	1.76	(0.98, 3.16)	22.86	0.71	(0.38, 1.34)
<200,000	27.06	22.64	0.79	(0.41, 1.5)	23.61	0.83	(0.44, 1.58)	11.76	0.36	(0.17, 0.76) *	28.79	1.09	(0.59, 2.02)	62.86	4.56	(2.51, 8.31) *

Note: OR = odds ratio; CI = confidence interval. * Significant difference. ^a^ Percentage of male athletes who participated in team sport programs in 2016 and who grew up in each of the subdivisions of the 2014 census. ^b^ Israel population = percentage of males under the age of 14 in each of the subdivisions of the 2014 Israel Census.

**Table 4 sports-10-00059-t004:** Representation of the Israeli population, female individual sport athletes ^a^, and ORs and CIs across cities of different sizes.

City Size	Israel Pop ^b^	Gymnastics			Judo			Swimming			Tennis			Track and Field		
		%	OR	CI	%	OR	CI	%	OR	CI	%	OR	CI	%	OR	CI
>2000	8.57	60.87	16.59	(0.81, 37.2)	28.00	4.14	(1.81, 9.46) *	10.00	1.18	(0.45, 3.0)	5.77	0.65	(0.21, 1.95)	18.52	2.42	(1.02, 5.74) *
2000–50,000	34.9	19.57	0.45	(0.23, 0.86) *	34.00	0.96	(0.53, 1.72)	28.00	0.72	(0.39, 1.32)	32.69	0.90	(0.50, 1.62)	48.15	1.73	(0.98, 3.05)
50,000–200,000	29.48	6.52	0.16	(0.06, 0.41) *	20.00	0.59	(0.31, 1.14)	22.00	0.67	(0.35, 1.27)	21.15	0.64	(0.33, 1.22)	11.11	0.29	(0.14, 0.63) *
<200,000	27.06	13.04	0.40	(0.19, 0.83)	18.00	0.59	(0.30, 1.16)	40.00	1.79	(0.99, 3.2)	40.38	1.82	(1.0, 3.31)	22.22	0.77	(0.40, 1.46)

Note: OR = odds ratio; CI = confidence interval. * Significant difference. ^a^ Percentage of female athletes who participated in individual sport programs in 2016 and who grew up in each of the subdivisions of the 2014 census. ^b^ Israel population = percentage of females under the age of 14 in each of the subdivisions of the 2014 Israel Census.

**Table 5 sports-10-00059-t005:** Representation of the Israeli population, female team sport players ^a^, and ORs and CIs across cities of different sizes.

City Size	Israel Pop ^b^	Basketball			Soccer			Team Handball			Volleyball		
		%	OR	CI	%	OR	CI	%	OR	CI	%	OR	CI
>2000	8.57	9.21	1.08	(0.40, 2.86)	4.76	0.53	(0.16, 1.69)	0.00	0.00	-	20.69	2.78	(1.18, 6.51) *
2000–50,000	34.9	21.05	0.49	(0.26, 0.93) *	47.62	1.69	(0.96, 2.99)	5.71	0.11	(0.04, 0.28) *	34.48	0.98	(0.54, 1.75)
50,000–200,000	29.48	27.63	0.91	(0.49, 1.68)	14.29	0.39	(0.19, 0.80) *	48.57	2.25	(1.26, 4.04) *	27.59	0.91	(0.49, 1.68)
<200,000	27.06	17.24	1.96	(1.08, 3.54) *	33.33	1.34	(0.73, 2.47)	45.71	2.26	(1.25, 4.09) *	17.24	0.56	(0.28, 1.10)

Note: OR = odds ratio; CI = confidence interval. * Significant difference. ^a^ Percentage of female athletes who participated in team sport programs in 2016 and who grew up in each of the subdivisions of the 2014 census. ^b^ Israel population = percentage of females under the age of 14 in each of the subdivisions of the 2014 Israel Census.

**Table 6 sports-10-00059-t006:** Coaches’ BPE preliminary reflections.

Question:
Do You Think That the Athletes’ Places of Birth Affect Their Development?
	Individual Sports	Team Sports
Yes	85.5% (71)	81.3% (52)
No	14.5% (12)	18.8% (12)
If yes, how does the place of birth contribute to the athletes’ development?
Proximity to sport facilities	
Socioeconomic status of the living place	
Sport popularity

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
