# Peer review of "The Birthplace Effect in 14–18-Year-Old Athletes Participating in Competitive Individual and Team Sports"

_sports, 2022, doi:10.3390/sports10040059_

Round 1
Reviewer 1 Report
General Comments
Thank you for the opportunity to review this paper. The paper is well written and takes on a mixed method approach to the BPE. There are some interesting findings. However, I feel that greater rationale was needed for the study and I was confused around the BPE data and how this sports participation data can be used. Some further explanation / clarification may help. I hope these comments help in improving the study
Specific Comments
Introduction
- Line 108 – The study aim “is to assess the BPE in young Israeli athletes (ages = 14-18 yrs.) who participated in competitive sport programs”. Why is this study needed? Although the introduction overviews research on the BPE it doesn’t critique the current literature enough to rationalise why a study on 14-18 young athletes is needed.
- Line 136 – consistent with my comment above, why is coaches perceptions of BPE important?
Methods
- Can you provide more context on the participants? Is this every participant at this age for these sports or a select sample? If a select sample – why this sample?
- Line 224 – l am confused here. The likelihood of participating compared to what? Compared to not participating? How was non participation determined with such a small sample?
- Line 240 – I think there is more than just descriptive analysis within this section. You use thematic analysis here
Results
- Table 2 – so the table shows the % of participants from each sport that were born in a city size. So out of the 18 male Gymnasts (5, 6, 1, 6). This links to m above point – is this every male gymnast aged 14-18 years? If not can we really determine anything from such data?
Discussion
- You refer to talent development here. How are the findings related to talent development when they are participation data?
- “Therefore, the contribution of small places and cities of a medium size to the development of the young athletes in our study can presumably be explained by the instructional and social characteristics of these places, as discussed in previous BPE studies” – Add references
- Some good insights within the qualitative data
- I like the policy implications
Author Response
Please see the Word attachment

Reviewer 2 Report
You describe the purpose of the study to: (a) to assess the BPE in young Israeli athletes (ages = 14-18 yrs.), and (b) to collect preliminary data on how the coaches perceived this effect. l recommend that you adjust your research question to be more precisely and to include something about city size.
I would like you to discuss in more detail what the potential benefits really are for the different sports and city sizes. For the both the quantitative results and for the coaches perception of the BPE. In addition, how the coaches perception is in consistency with the quantitative results.
I do not think that what report in the result section about the second part of your research question is very informative (coaches perception of the BPE). Is it possible to extend this section with more detailed information?
In the summary I recommend that you sum up the results based on your research questions. What you write in the summery now is a recommendation for policymakers (not a summary of results or conclusion). I would suggest that you write it as a conclusion, and that you in addition insert a section in the discussion about practical implications of your findings. Like you write it now you might be considered to biased (to have a kind of “political intention”).
Author Response
Please see the Word attachment

Round 2
Reviewer 1 Report
Thank you for addressing my comments